# Accelerated versus Slow In Vitro Aging Methods and Their Impact on Universal Chromatic, Urethane-Based Composites

**DOI:** 10.3390/ma16062143

**Published:** 2023-03-07

**Authors:** Nicoleta Ilie

**Affiliations:** Department of Conservative Dentistry and Periodontology, University Hospital, Ludwig-Maximilians-University, Goethestr. 70, D-80336 Munich, Germany; nilie@dent.med.uni-muenchen.de

**Keywords:** aging, resin-based composites, dynamic-mechanical analysis, hardness, modulus, universal chromatic

## Abstract

Structural coloring of dental resin-based composites (RBC) is used to create universal chromatic materials designed to meet any aesthetic need, replacing the mixing and matching of multiple shades. The microstructural adjustments to create this desideratum involve nanoscale organic–inorganic core–shell structures with a particular arrangement. The generally higher polymer content associated with these structures compared to universal chromatic RBCs colored by pigments, which in their microstructure come close to regularly shaded RBCs, can influence the way the material ages. Accelerated and slow aging up to 1.2 years of immersion in artificial saliva at 37 °C were therefore compared in relation to their effects on the materials described above and in relation to the immersion conditions prescribed by standards. Quasi-static and viscoelastic parameters were assessed to quantify these effects by a depth-sensing indentation test equipped with a DMA module. The microstructure of the materials was characterized by scanning electron microscopy. The results convincingly show a differentiated influence of the aging protocol on the measured properties, which was more sensitively reflected in the viscoelastic behavior. Accelerated aging, previously associated with the clinical behavior of RBCs, shows a 2- to 10-fold greater effect compared to slow aging in artificial saliva of up to 1.2 years, highly dependent on the microstructure of the material.

## 1. Introduction

One of the latest trends in the development of light-cured resin-based composites (RBC) is the design of materials intended to replace all shades, which is why they are called universal chromatic RBCs. In fact, these materials are chemically and, in the vast majority, also structurally similar to shaded RBCs, which is clearly reflected in a similar mechanical behavior [1,2].

The universal chromatic concept responds to a wish from practitioners to simplify the restorative procedure by skipping the decision as to which material shade should be selected to match the individual patient’s situation. The materials should adapt aesthetically to every situation and every patient [3,4]. This desiderate was only partially fulfilled as an improved blending effect to the environment was undeniably demonstrated [5,6,7], but the clinical reality was sobering, revealing a limitation of the blending effect to smaller restoration size [8,9,10].

In principle, two different technological directions can be identified in the development of universal chromatic RBCs. On the one hand, pigments are simply reduced or left out, whereby the microstructural features of the shaded materials are retained [1,2]. The absence of pigments, which absorb light to produce the intended shade, allows for the design of a more translucent material that naturally blends better with its surroundings. The second line of development uses more sophisticated technology to create distinct microstructures capable of selective light reflection [11], and is defined as structure-induced coloring.

Structure-induced coloring has hitherto been carried out in only one commercially available RBC. For this purpose, a nanoscale core–shell structure based on 260 nm large, polymer-coated silica and zirconium oxide particles were created. These coated nanoparticles are arranged to form larger, micron-sized spherical structures [12] but are also filling the spaces between the spherical structures, thus creating a multiscale organization of equidistant nano particles [1,2]. Because the polymer matrix that serves as the shell of the nanoscale core–shell structure is large, the total amount of inorganic silica and zirconia filler is lower compared to the inorganic filler amount of many regular RBCs. The proportion of inorganic filler has a decisive influence on the mechanical properties, in particular the modulus of elasticity [13], a fact that is reflected in the tendentially poorer mechanical properties of the structurally colored RBC [1,2].

Since the polymer part of an RBC and the filler–matrix interface are the ones that are prone to hydrolytic degradation [14], the structural-colored RBCs need to be carefully analyzed from this point of view to predicting clinical performance in the absence of relevant clinical studies. So far, a 2-year clinical study was published that evaluated the performance of the structure- and pigment-induced colored universal chromatic RBCs used in anterior veneer and diastema restorations. Within the limited observation time, the materials were defined as successful in terms of their clinical performance and color match [14]. When data from clinical studies are rare, one way to get closer to the clinical performance of a material is to subject it to artificial aging. This is either immersion in different media, mostly distilled water and artificial saliva [15,16], or additional thermal, mechanical, and biological aging [16,17]. The shortcoming here is that none of these methods are standardized and a correlation with clinical performance to assess whether the aging method can efficiently mimic clinical behavior has rarely been systematically performed. One of these is an aging protocol that ended with additional aging in a 75% ethanol/water solution, indicating a correlation between flexural strength and materials’ clinical performance [18]. The few published studies on structurally colored materials use either 3-month and 6-month water or artificial saliva immersion at 37 °C, or accelerated thermal aging (10,000 cycles, 5/55 °C) [19]. The results show little to no deterioration in the mechanical properties compared to values measured 24 h post-polymerization. In another train of thought, the structurally and pigment-colored universal chromatic RBCs have so far been shown to be non-toxic [12].

The study aims, therefore, to compare the effect of slow and accelerated aging protocols on the elasto-plastic and visco-elastic behavior of universal chromatic RBCs, either structurally or pigment colored. In addition to the coloring technology used, the focus when selecting the materials was set on their polymer matrix, which is Bis-GMA (bisphenol A glycol dimethacrylate)—free and preponderantly UDMA (urethane dimethacrylate) or UDMA derivatives (e.g., TCD-urethane: 2-propenoic acid, (octahydro-4,7 methano-1H-indene-5-diyl) bis (methyleneiminocarbonyloxy-2,1-ethanediyl) ester) based.

The null hypotheses tested are that the elasto-plastic and viscoelastic material behavior is independent of (a) type of RBC; (b) aging protocol: 24 h immersion in distilled water, slow (6 months and 1.2 years immersion in artificial saliva) or accelerated (additional immersion in ethanol solution) aging; and (c) frequency used (0.5 to 5 Hz).

## 2. Materials and Methods

### 2.1. Materials and Aging Methods

Three universal chromatic RBCs with a polymer matrix based on UDMA (Table 1) were exposed to four different aging protocols (Figure 1). A 24 h post-polymerization immersion in distilled water (24 h) served as reference. Thereafter, the samples were subjected to a 6-month immersion (6 M) in artificial saliva (pH 6.9; 1000 mL: 1.2 g potassium chloride, 0.84 g sodium chloride, 0.26 g dipotassium phosphate, 0.14 g calcium chloride dehydrate) with the medium being changed weekly. The 6 M aging was followed by the third and fourth aging methods, which consist of either additional accelerated aging that involved a 3-day immersion in a 75% ethanol—distilled water solution (6 M + ETOH) or prolonged immersion in artificial saliva for another 6 months, which corresponds to a total immersion time of 1.2 years (1.2 Y). All immersions occurred at 37 °C and dark environmental in an incubator. 

The influence of the aging protocol was quantified by quasi-static and viscoelastic analysis evaluated in an instrumented indentation test with a DMA (dynamic mechanical analysis) module.

### 2.2. Methods

(a)Specimen preparation

A number of 72 rectangular specimens (*n* = 24 per RBC) were prepared in white polyoxymethylene molds and cured with a blue LED (light-emitting diode) LCU (light curing unit) (Bluephase^®^ Style, Ivoclar Vivadent, Schaan, Liechtenstein, irradiance = 1400 mW/cm^2^). The sample preparation and curing were carried out according to ISO 4049:2019 [20]. For this purpose, a mold with an internal dimension of 2 mm × 2 mm × 8 mm was filled with the RBC paste, while two glass plates with polyacetate strips in between were used to compress the material. The specimens were polymerized from above and below for 20 s overlapping. After demolding, all RBC samples were immersed in distiller water at 37 °C for 24 h, as specified in ISO 4049:2019. A quarter (*n* = 18) of the test specimens (*n* = 6 per RBC) were then removed and tested as a reference group (24 h), while the remaining samples were further stored in artificial saliva at 37 °C for 6 months. At the end of the immersion period, one-third of the specimens were tested (6 M) and the rest were divided into two groups and subjected to further aging. One group aged accelerated by immersion in a solution consisting of a mixture of 75% absolute ethanol and 25% distilled water (6 M + ETOH) for 3 days, while the other group aged slowly by immersion in artificial saliva for additional 6 months (1.2 Y). Previous to measurements, specimens were wet-ground with silicon carbide abrasive paper (grit p1200, p2500, and p4000, LECO Corporation, Saint Joseph, MI, USA) and polished with a diamond suspension (mean grain size: 1 µm) for 2–3 min, until the surface was shiny (automatic grinding machine EXAKT 400CS Micro Grinding System, EXAKT Technologies Inc., Oklahoma City, OK, USA).

(b) Instrumented indentation test (IIT)

*Quasi-Static Instrumented Indentation test:* The IIT was performed in quasi-static mode based on ISO 14577 [21] using an automated nano-indenter (Fischerscope^®^ HM2000, Helmut Fischer, Sindelfingen, Germany). The used indenter, a Vickers diamond, has the shape of an orthogonal pyramid with a square base and an angle α = 68° between the axis of the diamond pyramid and one of the faces. The quasi-static indentation test was performed force-controlled, with the test force increasing within 20 s, then being kept constant for 5 s, and then decreasing within 20 s at a constant speed from 0.4 mN to 1000 mN. The test included one randomly selected indentation per sample (*n* = 6 per RBC). Load (F) and indentation depth (h) of the indenter were continuously measured during this load–unload cycle, allowing the elastic and plastic deformation to be calculated. Part of the mechanical work W_total_ (=∫Fdh) is consumed during the indentation process as plastic deformation work W_plast_, while the rest is released as work of elastic recovery W_elastic_. The ratio of the elastic recovery indentation work (W_elast_) and the total mechanical indentation work (W_total_) (W_elast_/W_total_ = µ_IT_) is a prerequisite for the subsequent DMA test. The force–depth dependence during the load–unload cycle was used for further parameter calculations. The slope of the tangent of the indentation depth curve at maximum force was used to calculate the indentation modulus E_IT_. Furthermore, as a measure of the resistance to plastic deformation, the indentation hardness parameter, H_IT_, was calculated as the ratio between the applied load and the contact area (H_IT =_ F_max_/A_p_) [21]. 

*Dynamic mechanical analysis (DMA):* The DMA involved an indentation force with a maximum of 1000 mN, similar to the test above. When the maximum force was reached, an oscillating force of small magnitude (10 different frequencies in the range of 0.5–5 Hz) was superimposed. The oscillation amplitude was set to five nm so that the sample deformation stayed within the linear viscoelastic regime. Six randomly chosen indentations have been performed per specimen, amounting to thirty-six individual measurements per RBC brand. Ten measurements were carried out per frequency (0.5 Hz; 0.7 Hz; 0.9 Hz; 1.1 Hz; 1.4 Hz; 1.8 Hz; 2.3 Hz; 3.0 Hz; 3.9 Hz; 5.0 Hz) and indentation.

At each frequency used, the force oscillation creates oscillations on the displacement signal with a phase angle δ. These sinusoidal response signals were then broken down into a real part and an imaginary part representing the storage (E′) and loss modulus (E″), respectively. Here, E′ is a measure of the elastic response of the material, while E″ characterizes the viscous material behavior. The quotient E″/E′ is defined as the loss factor (tan δ) and is a measure of the damping behavior of the material.

The indentation hardness, H_IT_, was determined along with the above-described viscoelastic parameters. 

(c) Morphological analysis of the filler (scanning electron microscopy, SEM)

The morphological appearance of the filler systems in the examined materials was determined in the electron backscatter diffraction mode using scanning electron microscopy (Zeiss Supra 55VP, Carl Zeiss GmbH, Goettingen, Germany). For this purpose, an additional set of rectangular bars (*n* = 3 per RBC) was obtained as described above and immersed in distilled water at 37 °C for 24 h prior to the surface preparation described above.

### 2.3. Statistical Analyses

The distribution of the variables was tested using the Shapiro–Wilk method. Since the variables were normally distributed, a parametric approach followed. A multi-factor analysis of variance was applied to compare the parameters of interest (Martens *HM*, Vickers *HV* and indentation *H_IT_* hardness, indentation modulus *E_IT_*, elastic *W_elastic_* and plastic deformation work *W_plast_*_,_ storage modulus *E*′, loss modulus *E*″, loss factor *tan δ*), among analyzed materials and loading frequencies. The multivariate analysis (general linear model) evaluates the effect of the main parameters—*RBC, aging method*, and frequency—as well as their interaction terms on the measured parameters. The partial eta-squared statistic reports the practical significance of each term, based on the ratio of the variation attributed to the effect. Larger values of partial eta-squared (η_P_^2^) indicate a greater amount of variation accounted for by the model. In addition, one- and multiple-way analysis of variance (ANOVA) and Tukey honestly significant difference (HSD) *post hoc* test using an alpha risk set at 5% was used (SPSS Inc. Version 27.0, Chicago, IL, USA). 

## 3. Results

### 3.1. Quasi-Static Parameters

A multifactorial analysis identified a significant (*p* < 0.001) influence of both main factors—aging method and RBC—on the measured quasi-static parameters. In a head-to-head comparison of all measured parameters, the effect size of factor RBC was larger than the aging method (higher partial eta-squared values, η_P_^2^, Table 2). The combined effect of the aging method and RBC was significant for three out of six parameters. 

The measured parameters enable a clear classification of the tested RBCs. For hardness (HM and HV) and modulus of elasticity (E_IT_), the ranking is OC < VPO < VDO (Figure 2, Figure 3, Figure 4 and Figure 5), while for n_IT,_ W_e_, and W_t_, VPO and VDO are statistically similar and lower than OC.

On OC, the 24 h, 6 M, and 1.2 Y aging were statistically similar with respect to all quasi-static parameters, while a strong degradation occurred for additional alcohol immersion (6 M + ETOH). In contrast, the aging process, including the alcohol immersion, showed no significant effect on VPO. As with VDO, additional alcohol immersion slightly but significantly decreased HM and HV but not the other parameters. Finally, it should be noted that for all materials tested and for all parameters, the confidence interval for alcohol aging was larger than for other aging conditions.

### 3.2. Visco-Elastic Parameters

The results of the dynamic mechanical analysis indicate a significant (*p* < 0.001) and strong effect (Table 3) of all three main factors—aging method, RBC, and frequency—on the measured parameters. Similar to the quasi-static test, the effect strength of the RBC was stronger than the aging method (higher partial eta-squared values, η_P_^2^, Table 3). Frequency shows the strongest effect on the loss factor and the storage modulus. 

The variation of the measured parameters with the aging method and frequency is shown for the material OC as an example. The variation with frequency for H_IT_ was small (Figure 5), evidencing a slight increase up to 1.1 Hz. This pattern of variation was similar in all materials and for all aging methods. 

In OC, H_IT_ was statistically highest and similar for the 24 h and 6 M aging protocols (*p* = 0.988). A total of 1.2 Y aging reduced the values only slightly but significantly, while immersing the specimens in alcohol after 6 M aging reduced the hardness drastically.

Similar to the quasi-static results, note the greater variability in the data after accelerated aging (6 M + ETOH).

The pattern of variation of storage modulus with frequency (Figure 6) shows the highest values at the lowest frequencies, followed by a steady decrease up to 1.1 Hz and then a plateau, as frequencies increased. Again, this pattern of variation was similar for all materials and for all aging conditions. In contrast to the H_IT_ results, E_IT_ in OC was significantly highest at 24 h aging and then slightly decreased at 6 M and 1.2 Y storage in artificial saliva. The last two groups were statistically similar (*p* = 0.178). A strong decrease is observed on immersion in ETOH. 

The loss modulus (Figure 7) shows an exponential variation within the tested frequency range with statistically similar values for 24 h, 6 M, and 1.2 Y (*p* = 0.158) aging and statistically significantly higher values after additional ETOH immersion at lower frequencies (up to 1.4 Hz).

Similarly, the loss factor (Figure 8) decreases exponentially with frequency. No distinction was made between 24 h, 6 M, and 1.2 Y (*p* = 0.495), while aging in alcohol increases significantly values at all frequencies, the most, the lower the frequency.

The material comparison and the effect of the aging process are presented more decidedly below using an exemplary frequency (1.1 Hz). Overall, the effect size of the parameter RBC is higher than the aging method (higher η_P_^2^ values, Table 4) thus confirming the relation calculated previously over all frequencies, as was summarized in Table 3. 

The indentation hardness (Figure 9) was significantly the highest for VDO, followed by VPO and OC. The effect of the aging method was similar for VDO and VPO in that longer storage in artificial saliva lowered the 24 h results significantly, while there was no difference between 6 months or 1.2 years immersion in artificial saliva. Accelerated aging in alcohol significantly reduces the values. In contrast, OC was not significantly affected by immersion in artificial saliva for up to 1.2 years. However, additional immersion in alcohol greatly reduces the values.

The results for the storage modulus are summarized in Figure 10 and show a similar statistical ranking and dependency on the aging process as described above for the indentation hardness.

The loss factor (Figure 11) increases in the range VDO > VPO > OC and corresponds to the described statistical comparison of the influence of the aging method. A very high increase due to the alcohol storage can be noticed for OC, which represents a decrease in properties to almost 50% of the values measured for the other aging methods, including the reference.

The structural appearance of the filler system in OC shows nanoparticles clustered in larger spheres of different micron size with some inherent porosity. In contrast, the fillers in VDO and VPO are irregular, crushed, solid glass fillers up to 5–6 µm in size and quite similar in appearance in both materials (Figure 12).

## 4. Discussion

Dental restorative materials are exposed to a challenging environment in the oral cavity over a long period of time, which can alter their initial, targeted, properties. Since the materials studied are polymer-based composites that function below their glass transition temperature, the polymer part of the material is subject to physical aging, also defined as structural recovery [22]. Due to the rapid light-curing process, the polymer is normally in a non-equilibrium thermodynamic state while the polymer chains undergo thorough conformational changes en route to a thermodynamically stable state. These events at the molecular level involve modifications at the macroscopic level, which are manifested as changes in physical properties [22]. In addition to physical aging, RBCs undergo further changes when exposed to saliva and its enzymatic compounds, dietary components, or changes in temperature and pH. The result is hydrolytic degradation or, in the presence of enzymes, biodegradation [23,24,25]. Since the degradation affects a number of functional groups in the polymer such as esters, ethers, urethanes, and amides that can be cleaved by hydrolysis [23], the chemical specifics of the monomers have an impact. To reduce the influence of the chemical composition, the RBC selection was restricted to materials containing UDMA, TCD-urethane, and TEGDMA as the dilute monomer (Table 1). In addition, hydrolytic and biodegradation are not limited to the polymer matrix since the silane that binds the filler into the organic matrix and its degree of adhesion provides one more link that is susceptible to chemical attack and degradation [26]. 

The main function of fillers in resin composites is to bear the load and thereby increase several key mechanical properties such as strength and modulus of elasticity [13,27]. In addition to the particle-matrix adhesion strength, the amount but also the shape of the fillers is of great importance, since irregularly shaped and larger fillers have a higher potential for improving mechanical properties compared to round and smaller fillers [27]. From this point of view, the filler systems, the number of fillers and the way the fillers are incorporated into the organic matrix are quite different in OC compared to VDO and VPO. OC is a core–shell hybrid organic–inorganic RBC consisting of two distinct components, a centered nano-sized inorganic core, either silica or zirconia, and a surrounding polymeric shell. The size of the inorganic fillers and shell, their nature, and three-dimensional distribution were adjusted so that they can produce a red–yellow structural color [28] when ambient light passes through the material, to match the natural color of human teeth. In fact, the inorganic filler appears to be of very regular size, confirming the manufacturer’s dimensions of 260 nm (Figure 12). Whether there is silanization of the nano-core–shell structures within the larger spherical formations, or of the spheres within the organic matrix, has not been described so far. It should be noted that the total amount of inorganic filler is lower compared to the stated total amount of filler since the polymer shell need to be excluded. The lower content of inorganic fillers was directly reflected in the measured micromechanical parameters such as the indentation modulus and hardness, thus clearly confirming previous studies [1,2]. The SEM images (Figure 12) taken in electron backscatter diffraction mode on non-sputtered samples, evidenced a difference between the white zirconia fillers and the light gray silica fillers, as the backscatter mode is sensitive to an element’s atomic number, generating a whiter appearance of structures containing elements with higher atomic numbers. In addition, the images allow differentiation of the voids inherent in the spheres that group the nanofillers, as these areas appear much darker than the polymer matrix. In contrast to the structure described for OC, the Ba-Al-B-F-Si glass fillers in VPO and VDO exhibit irregular shapes and a multimodal filler distribution (Figure 12). The higher number of inorganic fillers in VDO and VPO compared to OC, but also the shape of the filler, induced higher micromechanical properties in the former materials. The methodology used allows for a clear distinction between VDO and VPO, which are quite similar in their microstructure and filler size. However, the lower amount of filler and the addition of a small amount (1% by volume) of pre-polymerized fillers in VPO was registered as a significant decrease in modulus and hardness, confirming previous studies [29]. Since VDO and VPO are chemically and structurally identical to the regularly shaded RBCs Venus Diamond and Venus Pearl, both the viscoelastic and elastic–plastic parameters are statistically comparable, confirming previous studies [1,2]. Although the measured mechanical parameters were in comparison weaker in OC, while the damping behavior was better due to the higher polymer content, all three materials fit well into the regular shaded light-cured RBC materials category, when analyzed under identical storage and methodological setup conditions [1,2].

Most laboratory aging experiments are performed with the goal of understanding the performance and degradation mechanisms of RBCs in a clinical setting. Accelerated aging is often used to shorten the duration of an experiment and is frequently achieved by exposing samples to extremely harsh environments, usually mechanical and/or hydrothermal stress [17], high UV or visible light exposure [30], solvents with plasticizing effects [15,16,31] or biofilms [32]. As with artificial aging, the correlation of the results with behavior under clinical service conditions may not be straightforward. The aging protocols selected in the current study included slow aging under clinically relevant conditions and an accelerated aging protocol that has been shown to be efficient in mimicking clinical material behavior [18]. Slow aging involves immersion in artificial saliva at 37 °C for up to 1.2 years. These conditions are based on the literature data that indicate that saturation of typical dental RBCs in artificial saliva or water occurs over a 4–5 month period, underscoring the importance of examining the mechanical properties in a composite that has been aged for an extended period of time [16]. The accelerated aging protocol added ethanol aging to 6 months of storage under the above conditions. A 24 h storage at 37 °C and distilled water was chosen as the baseline for the test, as these are the conditions specified in the most widely used standard for testing dental RBCs (ISO 4049 [20]), allowing comparison with various other available results based on the standard protocol.

In addition to the laboratory simulation of clinical aging, the search for a qualified parameter to precisely quantify the often very subtle age-related material changes remains a challenge. Micromechanical parameters such as hardness, expressed either as plastic (Vickers hardness) or as elastic–plastic deformation (Universal or Martens hardness), but also the indentation modulus measured in a quasi-static approach, as performed in the present study, proved to be more responsive parameters in detecting aging effects in RBCs than the flexural strength [29]. In no way does this statement allow diminishing the importance of flexural strength testing, but rather emphasizes the importance of additional Weibull analysis since it has been shown that the reliability of the materials, expressed by the Weibull parameter, is a similarly valuable and sensible predictor of the aging process as the micromechanical parameters [2]. If the materials are subjected to an oscillating load with different frequencies in addition to the quasi-static approach, the measured viscoelastic parameters turn out to be even more discriminating for the aging process, since they can already differentiate small changes in the materials that remained undetected in the quasi-static analysis. This was particularly evident for the small effects caused by immersion in artificial saliva in relation to the reference test performed at 24 h immersion. 

The static and dynamic complementary analyses evidenced that the way a material age depends more on the material itself than on the aging protocol. The degradation effect of an aging protocol thereby cannot be described unambiguously without allowance for the material structure peculiarities. In this context, the structurally colored universal chromatic RBC, OC, was much more strongly affected by accelerated aging in alcohol solution than VDO and VPO, the two pigment-colored materials. This needs to be related to the distinct microstructure in OC described above. Due to the microstructure, the material has an extremely large filler/matrix interface ratio, which is not only susceptible to hydrolytic degradation [26] and higher water uptake [33] but also offers a large stress-dissipation surface, since stress can be dissipated at such interfaces through friction [34]. This aspect is reflected very clearly in the largest loss factor measured in OC versus VDO and VPO for each of the aging protocols tested, particularly during the accelerated aging protocol. In addition to the large stress-dissipating interface, the damping behavior in RBCs is determined by the polymer content and the filler amount and size [35], which is favored with OC due to the higher polymer content and lower inorganic filler amount and size. This line of reasoning is valid also in the comparison of VDO vs. VPO, while the materials evidenced fewer structural differences. The damping ability was indeed higher in VPO, the material with the lower filler amount in this direct comparison. As observed in all RBCs analyzed so far, the hardness and indentations modulus on the one side and the damping ability on the other are mutually exclusive [35].

The larger surface area-to-volume ratio associated with small fillers was demonstrated to increase water uptake and consequent degradation of the filler/matrix interface [26], thereby compromising long-term mechanical properties. Interestingly, the degradation of OC in the slow aging protocol up to 1.2 years of immersion in artificial saliva was low, in the range of approx. 2–3% based on 24 h storage and hardly noticeable in the statistics. On the other hand, the additional storage in alcohol solution, which represents a quite aggressive aging condition, has a strong and differentiated effect. It causes a decrease in properties of about 30%, which, to quantify this effect, means a 10 times higher impact on the measured properties compared to the slow aging effect. Plausible in this context would be the significantly higher proportion of the organic matrix prone to be plasticized when immersed in the solvent. In contrast, the difference between the two aging methods was significantly smaller for VDO and VPO. Although slow aging had a significantly greater effect than that observed with OC, the additional aging in alcohol showed only a slight decline in properties. In this respect, the effect of accelerated aging was 2× greater than slow aging. The amount of filler and the interface are of great importance to understanding degradation, but the fact that VDO and VPO also contain in their matrix a solvent-stable monomer such as TCD-urethane, which, due to its higher viscosity, requires less TEGDMA, may be an argument that needs to be considered.

In order to emphasize the strengths and weaknesses of the study, it can be stated that the methods used, primarily the dynamic mechanical analysis, are demonstrably able to sensitively record small changes in the behavior of the analyzed materials during aging. In addition, data show that the mechanical properties of materials depend differently, thus material specifically, on the type of aging, making it difficult to define a general valid aging protocol.

## 5. Conclusions

The material has a stronger influence on the measured parameters than the aging method. Accelerated aging shows a 2 to 10 times greater effect compared to slow aging in artificial saliva of up to 1.2 years. The IIT-derived mechanical properties and particular DMA proved to be suitable and sensitive to quantify the effect of aging. Altogether, our results provide compelling evidence for a differentiated influence of the aging protocol on the measured properties.

## Figures and Tables

**Figure 1 materials-16-02143-f001:**
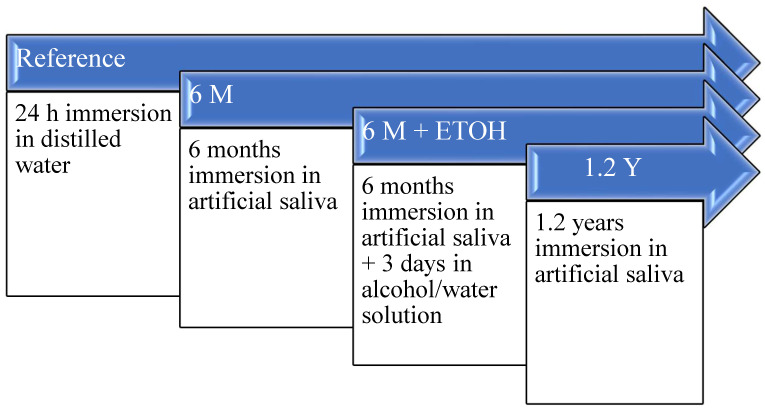
Overview of the four aging methods.

**Figure 2 materials-16-02143-f002:**
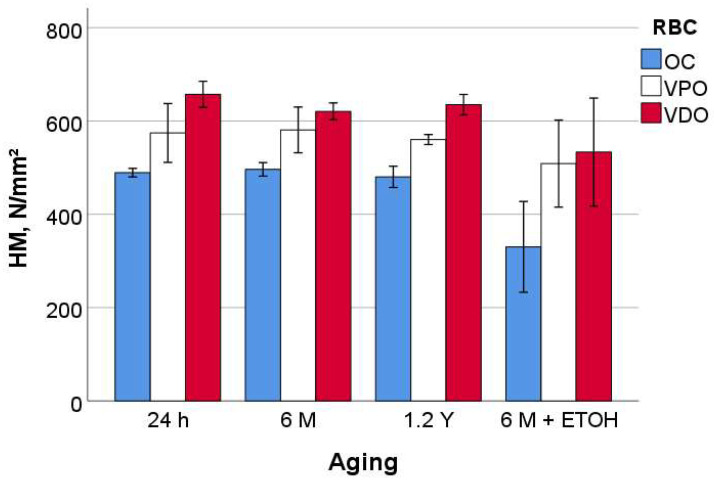
Martens hardness (HM) as a function of RBC and aging method (mean values with 95% confidence interval).

**Figure 3 materials-16-02143-f003:**
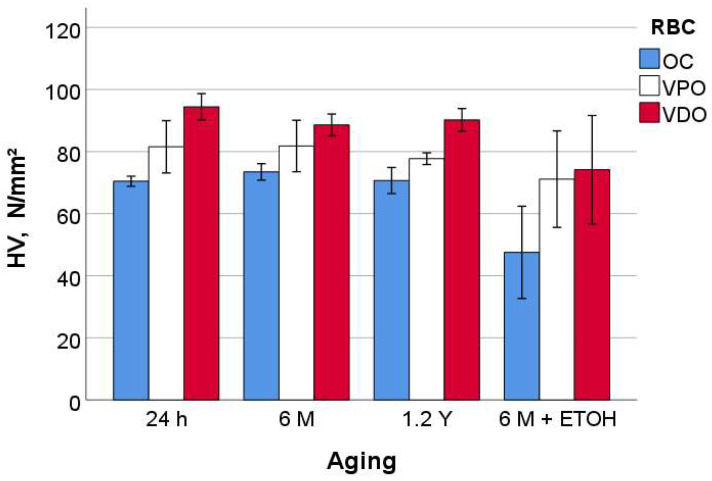
Vickers hardness (HV) as a function of RBC and aging method (mean values with 95% confidence interval).

**Figure 4 materials-16-02143-f004:**
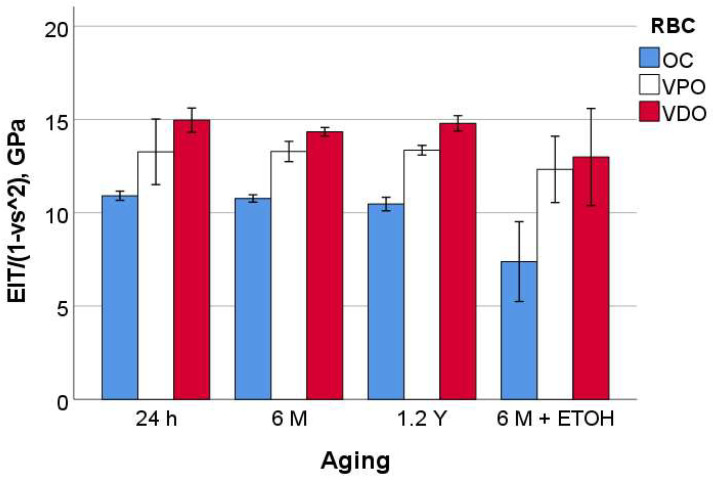
Indentation modulus (E_IT_) as a function of RBC and aging method (mean values with 95% confidence interval).

**Figure 5 materials-16-02143-f005:**
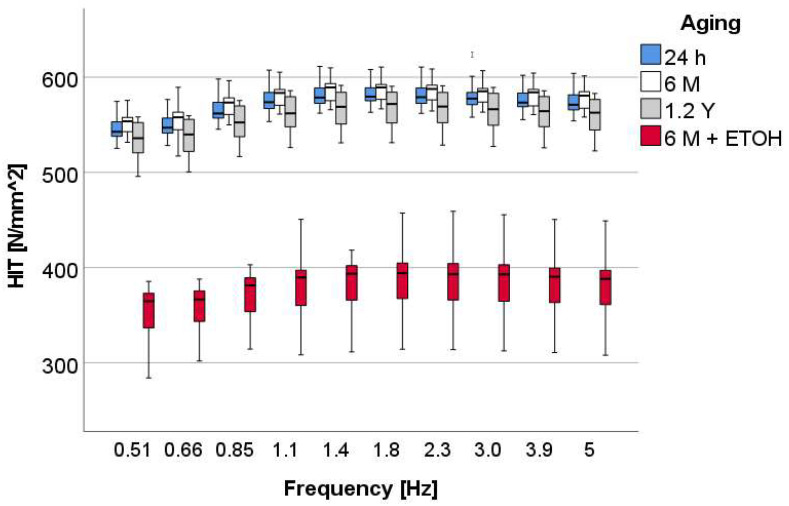
Indentation hardness (H_IT_) as a function of frequency and aging method, exemplified for the material OC.

**Figure 6 materials-16-02143-f006:**
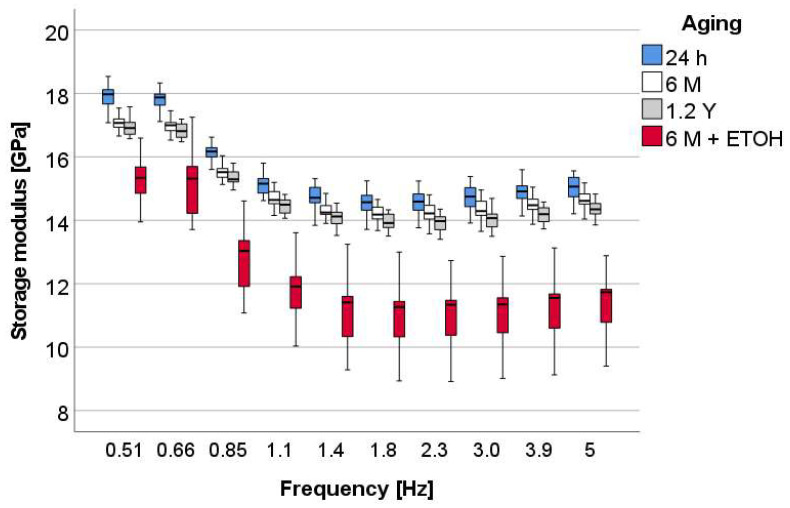
Storage modulus (E′) as a function of frequency and aging method, exemplified for the material OC.

**Figure 7 materials-16-02143-f007:**
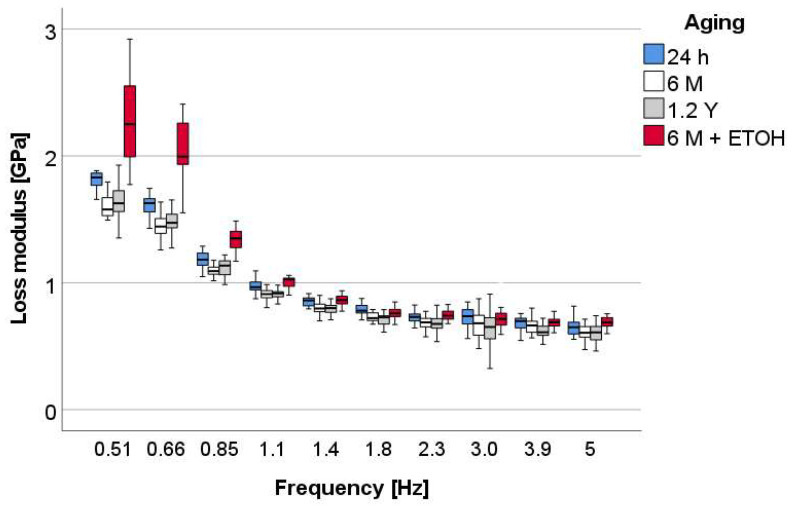
Loss modulus (E″) as a function of frequency and aging method, exemplified for the material OC.

**Figure 8 materials-16-02143-f008:**
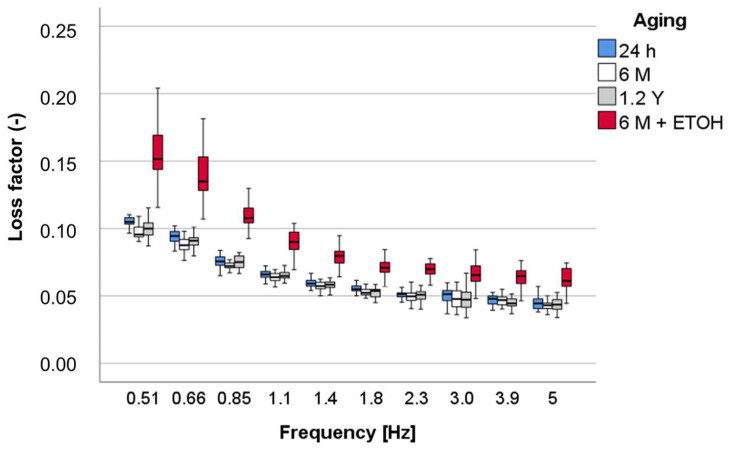
Loss factor (tan δ) as a function of frequency and aging method, exemplified for the material OC.

**Figure 9 materials-16-02143-f009:**
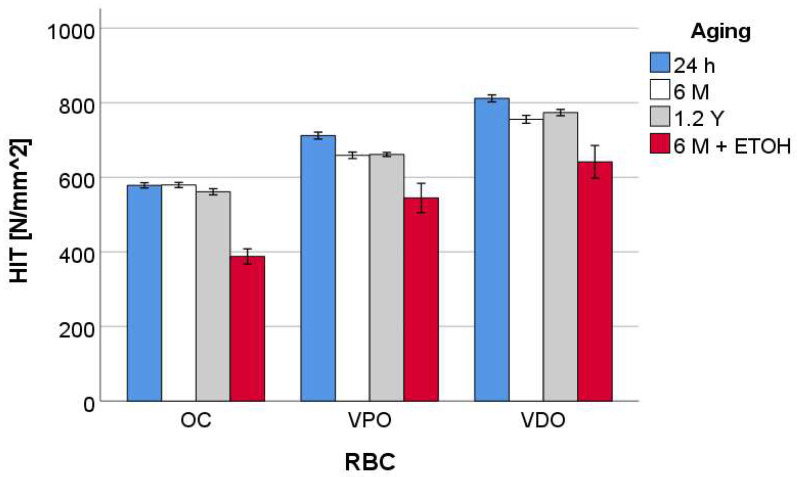
Indentation hardness (H_IT_) as a function of RBC and aging method, exemplified for a frequency of 1 Hz

**Figure 10 materials-16-02143-f010:**
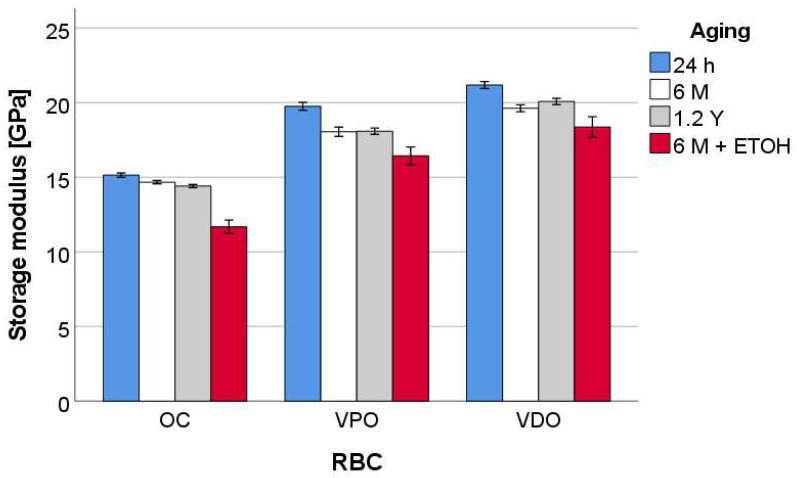
Storage modulus (E′) as a function of RBC and aging method, exemplified for a frequency of 1 Hz.

**Figure 11 materials-16-02143-f011:**
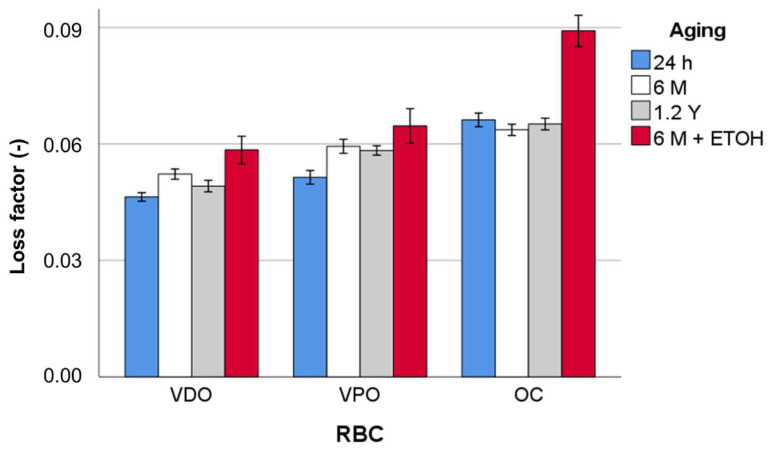
Loss factor (tan δ) as a function of RBC and aging method, exemplified for a frequency of 1 Hz.

**Figure 12 materials-16-02143-f012:**

SEM images, electron backscatter diffraction mode. Structural appearance of the filler system in OC = Omnichroma; VDO = Venus Diamond One, and VPO = Venus Pearl One in 3000× magnifications (scale = 2 μm).

**Table 1 materials-16-02143-t001:** Analyzed universal chromatic RBCs, abbreviation code, brand, LOT, and composition, as indicated by the manufacturer.

Code	Material Manufacturer	LOT	Monomer	Filler	
				Composition/Size	wt/vol%
OC	Omnichroma Tokuyama Dental	0551	UDMA, TEGDMA	SiO_2_, ZrO_2_ 260 nm	79/68
VDO	Venus Diamond ONE Heraeus Kulzer	VP221019	TCD-urethane UDMA TEGDMA	Ba-Al-B-F-Si glass, SiO_2_ 5 nm–20 μm d_50_ = 1.8 µm	81/64
VPO	Venus Pearl ONE Heraeus Kulzer	VP211019	TCD-urethane UDMA TEGDMA	Ba-Al-B-F-Si glass PPF, SiO_2_ 5 nm–5 μm d_50_ = 1.3 µm	75/59_t_ (58)_i_

Abbreviations: UDMA = urethane dimethacrylate; TCD-urethane = 2-propenoic acid, (octahydro-4,7 methano-1H-indene-5-diyl) bis (methyleneiminocarbonyloxy-2,1-ethanediyl) ester; TEGDMA = triethylene glycol dimethacrylate; PPF = pre-polymerized filler; SiO_2_ = silicon oxide (silica); ZrO_2_ = zirconium oxide (zirconia); Ba-Al-B-F-Si glass = BaO-Al_2_O_3_-B_2_O_3_-F-SiO_2_ (barium aluminum boro fluor silicate) glass; d_50_= mean particle size; wt%= percent by weight; vol%= percent by volume; subscript t = total filler amount; subscript i = inorganic filler amount.

**Table 2 materials-16-02143-t002:** Multivariate analysis (general linear model) and the effect of the main factors—aging method and RBC on the tested parameters. Partial eta-squared (η_P_^2^) values are indicated (n.s. = not significant).

	HM N/mm^2^	HV N/mm^2^	E_IT_/(1 − vs^2^) GPa	n_IT_ %	W_e_ µJ	W_t_ µJ
Aging method	0.561	0.556	0.484	0.049	0.482	0.376
RBC	0.722	0.643	0.808	0.635	0.801	0.590
Aging × RBC	n.s.	n.s.	n.s.	0.230	0.433	0.302

**Table 3 materials-16-02143-t003:** Multivariate analysis (general linear model) and the effect strength of the main factors—aging method, RBC, and frequency—on the parameters tested. All effects were statistically significant (*p* < 0.001); partial eta-squared (η_P_^2^) values are given.

	H_IT_ N/mm^2^	Loss Modulus GPa	Loss Factor (-)	Storage Modulus GPa
Aging method	0.638	0.684	0.159	0.498
RBC	0.896	0.793	0.082	0.591
Frequency	0.685	0.066	0.935	0.870

**Table 4 materials-16-02143-t004:** Multivariate analysis (general linear model) and the effect strength of the main factors—aging method and RBC—and their combined effect on the parameters tested and a frequency of 1.1 Hz. All effects were statistically significant (*p* < 0.001); partial eta-squared (η_P_^2^) values are given.

	H_IT_ N/mm^2^	Loss Modulus GPa	Loss Factor (-)	Storage Modulus GPa
Aging method	0.661	0.684	0.101	0.541
RBC	0.898	0.791	0.142	0.668
Aging × RBC	0.116	0.085	0.130	0.284

## Data Availability

Data is available upon personal request.

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
