# Peer review of "Accelerated versus Slow In Vitro Aging Methods and Their Impact on Universal Chromatic, Urethane-Based Composites"

_materials, 2023, doi:10.3390/ma16062143_

Round 1

Reviewer 1 Report

 A paper was published having title "Universal Chromatic Resin-Based Composites: Aging Behavior Quantified by Quasi-Static and Viscoelastic Behavior Analysis" by the same author is quiet similar with this manuscript only one variable has been changed. 

Author should focus on the discussion part of the study as the author discuss their results but didnt discuss the results with other similar studies.

Should include the weakness strength and limitations of the study.

References should be change according to the MDPI guidelines

Author Response

All comments to the corresponding author have been addressed independently below. The authors’ rebuttal is always in BLUE and where changes have been added to the revised manuscript in light of the reviewer's comments these are presented in RED.

The author would firstly like to thank the reviewers for taking the time to read and critically appraise the manuscript and secondly to thank the reviewers for their positive constructive comments in improving the work.

Reviewer 1

Comments and Suggestions for Authors

 A paper was published having title "Universal Chromatic Resin-Based Composites: Aging Behavior Quantified by Quasi-Static and Viscoelastic Behavior Analysis" by the same author is quiet similar with this manuscript only one variable has been changed. 

Author’s response:  I strongly disagree with the reviewer on this point. The present work analyzed the influence of aging by comparing how slow and accelerated aging protocols affect the elasto-plastic and viscoelastic material behavior. Comparisons of this kind are missing in the literature and are discussed controversially due to the lack of reliable data. The paper you mentioned deals with early material stages and a totally different study design. With the topic of universal chromatic RBCs now becoming the focus of interest for clinicians and researchers, the novelty of the materials means that they have been less studied in vitro and even less analyzed in vivo. The number of materials of this type is small, there is only one to the structurally colored materials to be more precise (OC), which understandably then leads to the materials being incorporated into many study designs. As described in the introduction, there is only one clinical study on the material category. Therefore, there is a great need to simulate their long-term behavior in the laboratory to give clinicians confidence in using this novel category of materials. Regarding the methods used, which are indeed similar to our previous study, which is cited in the present study, that focused on the macro-mechanical properties and fractography, please keep in mind that we need reliable methods to quantify effects and compare results measured under different conditions to understand material behavior and predict how they may behave clinically.

Author should focus on the discussion part of the study as the author discuss their results but didnt discuss the results with other similar studies.

Author’s response:  I extended the discussion following the reviewer's suggestion.

Should include the weakness strength and limitations of the study.

Author’s response: Thanks for pointing out this aspect.  At the end of the discussion, the strengths and weaknesses of the study are now presented.

References should be change according to the MDPI guidelines

Author’s response:  I checked the references as recommended by the reviewer.  Thank you. I apologize for the mistake.

There is kind of a tendency to self-citation (9 articles !). By the way I suggest some minor adjustements to improve the article for the readers.

Author’s response:  indeed, I didn't notice the high amount of self-citations! I reduced them in the revision. However, since we research RBCs very intensively and are mostly among the first to report on new trends and new materials, in addition to having built up the world's largest database on RBCs which allows for more complex data analysis than studies examining 3-4 materials, I inevitably have to refer to such articles to keep up with the current literature. I apologize for the excess.

Reviewer 2 Report

1. What is clearly noticed is that the Author included 10 of self-citations within 32 positions of references. They should either be reduced or the list of references widened to at least 50 positions (without more self citations), see the examples of papers:

Grzebieluch, W.; Kowalewski, P.; Grygier, D.; Rutkowska-Gorczyca, M.; Kozakiewicz, M.; Jurczyszyn, K. Printable and Machinable Dental Restorative Composites for CAD/CAM Application—Comparison of Mechanical Properties, Fractographic, Texture and Fractal Dimension Analysis. Materials 202114, 4919. https://doi.org/10.3390/ma14174919

(the interesting paper concerning fractal analysis of the presented by the Author dental materials)

Kul E, Abdulrahim R, Bayındır F, Matori KA, Gül P. Evaluation of the color stability of temporary materials produced with CAD/CAM. Dent Med Probl. 2021;58(2):187–191. doi:10.17219/dmp/126745

(the paper reflects directly to the topic Author touches)

Paradowska-Stolarz, A.; Malysa, A.; Mikulewicz, M. Comparison of the Compression and Tensile Modulus of Two Chosen Resins Used in Dentistry for 3D Printing. Materials 202215, 8956. https://doi.org/10.3390/ma15248956

(the paper presents mechanical features, properties of the chosen resins printed with use of 3-D technology)

All the presented papers refer to CAD-CAM technique and are the examples of papers that could be used for widening the literature. Please, find more - as mentioned 50 in total in this case so that not more than 20% of papers are the Authors.

2. In the text fig. 1 goes after table 1. Please, change that with graphs as well

3. Line 102 - please explain what you mean by "dark environment" - where were the specimens kept? Incubator? Specify when was the research done, eg. April -June 2022. This refers to all the descriptions. Please, specify at least year when the research was conducted (or years) 

4. The limitations of the study ar missing

5. Because the research is an in vitro study, it should be mentioned in the title.

Beside that, the paper is really well prepared. Please, correct the flaws I pointed out before further steps of publishing the paper.

Author Response

All comments to the corresponding author have been addressed independently below. The authors’ rebuttal is always in BLUE and where changes have been added to the revised manuscript in light of the reviewer's comments these are presented in RED.

The author would firstly like to thank the reviewers for taking the time to read and critically appraise the manuscript and secondly to thank the reviewers for their positive constructive comments in improving the work.

Reviewer 2

Comments and Suggestions for Authors

  1. What is clearly noticed is that the Author included 10 of self-citations within 32 positions of references. They should either be reduced or the list of references widened to at least 50 positions (without more self citations), see the examples of papers:

- Grzebieluch, W.; Kowalewski, P.; Grygier, D.; Rutkowska-Gorczyca, M.; Kozakiewicz, M.; Jurczyszyn, K. Printable and Machinable Dental Restorative Composites for CAD/CAM Application—Comparison of Mechanical Properties, Fractographic, Texture and Fractal Dimension Analysis. Materials 202114, 4919. https://doi.org/10.3390/ma14174919

(the interesting paper concerning fractal analysis of the presented by the Author dental materials)

- Kul E, Abdulrahim R, Bayındır F, Matori KA, Gül P. Evaluation of the color stability of temporary materials produced with CAD/CAM. Dent Med Probl. 2021;58(2):187–191. doi:10.17219/dmp/126745

(the paper reflects directly to the topic Author touches)

- Paradowska-Stolarz, A.; Malysa, A.; Mikulewicz, M. Comparison of the Compression and Tensile Modulus of Two Chosen Resins Used in Dentistry for 3D Printing. Materials 202215, 8956. https://doi.org/10.3390/ma15248956

(the paper presents mechanical features, properties of the chosen resins printed with use of 3-D technology)

All the presented papers refer to CAD-CAM technique and are the examples of papers that could be used for widening the literature. Please, find more - as mentioned 50 in total in this case so that not more than 20% of papers are the Authors.

Author’s response:  indeed, I didn't notice the high amount of self-citations! I reduced them in the revision. However, since we research RBCs very intensively and are mostly among the first to report on new trends and new materials, in addition to having built up the world's largest database on RBCs which allows for more complex data analysis than studies examining 3-4 materials, I inevitably have to refer to such complex articles to keep up with the current literature. I apologize for the excess.

I would also like to thank you for the literature recommended to me, which I read carefully. As the reviewer also noted, the literature deals with CAD/CAM RBCs or 3D printed RBCs, which is not the subject of the paper. CAD/CAM RBCs are industrially hardened under different conditions, and thus present advantages and disadvantages that detract from the subject of the study. Besides using other types of materials, the recommended papers focuses on fractography and macroscopically measured parameters such as strength and modulus, and lesson the visco-elastic or elasto-plastic material behaviour.

In addition, I have expanded the literature to include topic-related works, in particular, related to aging methods. In fact, the literature list in original research papers should be kept as small and representative as possible, since it is the task of literature reviews to present a topic exhaustively.

  1. In the text fig. 1 goes after table 1. Please, change that with graphs as well

Author’s response: Thank you for this observation. Figures and tables have now been placed in the order in which they are cited in the text.

  1. Line 102 - please explain what you mean by "dark environment" - where were the specimens kept? Incubator? Specify when was the research done, eg. April -June 2022. This refers to all the descriptions. Please, specify at least year when the research was conducted (or years) 

Author’s response: yes, an incubator was meant here.  I added the information for more clarity. Regarding the question related to the period of time in which the study was performed, it was ca. august 2021- November 2022, as it involved a 1.2 years period of immersion. This type of information is never included in a paper and is in my view rather disturbing. As all graphical representations involved all storage conditions, repeating that the study was performed from august 2021- November 2022 ca. 12 times, would be unusual as well. 

  1. The limitations of the study ar missing

Author’s response: Thanks for pointing out this aspect.  At the end of the discussion, the strengths and weaknesses of the study are now presented.

  1. Because the research is an in vitro study, it should be mentioned in the title.

Author’s response: Thank you for the suggestion. The word in-vitro has been added to the title.

Beside that, the paper is really well prepared. Please, correct the flaws I pointed out before further steps of publishing the paper.

Author’s response: Thank you for your praise and the constructive evaluation of my work.

Round 2

Reviewer 2 Report

Thank you for the corrections. I find the article more appropriate to be published in that manner. Thank you